# Effects of Multidisciplinary Biopsychosocial Rehabilitation on Short-Term Pain and Disability in Chronic Low Back Pain: A Systematic Review with Network Meta-Analysis

**DOI:** 10.3390/jcm12237489

**Published:** 2023-12-04

**Authors:** Ivan Jurak, Kristina Delaš, Lana Erjavec, Janez Stare, Igor Locatelli

**Affiliations:** 1Department of Physiotherapy, University of Applied Health Sciences, 10000 Zagreb, Croatia; ivan.jurak@zvu.hr (I.J.);; 2Faculty of Medicine, University of Ljubljana, 1000 Ljubljana, Slovenia; 3PhysioPlus, 10000 Zagreb, Croatia; 4Faculty of Pharmacy, University of Ljubljana, 1000 Ljubljana, Slovenia

**Keywords:** chronic low back pain, multidisciplinary biopsychosocial rehabilitation, exercise therapy, network meta-analysis

## Abstract

Chronic low back pain (CLBP) is a significant public health issue, with prevalence intensifying due to an ageing global population, amassing approximately 619 million cases in 2020 and projected to escalate to 843 million by 2050. In this study, we analyzed the effects of multidisciplinary biopsychosocial rehabilitation (MBR) on pain and disability. To address this question, we conducted a PRISMA-guided systematic review and random-effect network meta-analysis on studies collected from six electronic databases. The network comprised diverse MBR modalities (behavioral, educational, and work conditioning) alongside exercise therapy (ET), minimal intervention, and usual care, with pain and disability as outcomes. Ninety-three studies were included, encompassing a total of 8059 participants. The NMA substantiated that both ET and MBR modalities were effective in alleviating CLBP, with education-oriented MBR emerging as the most efficacious for pain mitigation (MD = 18.29; 95% CI = 13.70; 22.89) and behavior-focused MBR being the most efficacious for disability reduction (SMD = 0.88; 95% CI = 0.46; 1.30). Nevertheless, the discerned differences amongst the treatments were minimal and uncertain, highlighting that no modality was definitively superior to the others. Given the intricate nature of CLBP, embodying various facets, our findings advocate for a combined therapeutic approach to optimize treatment efficacy.

## 1. Introduction

Chronic low back pain (CLBP) is characterized by pain and discomfort localized below the inferior margin of the 12th ribs and above the inferior gluteal folds, persisting for a duration of at least 12 weeks, usually without a specific cause of the pain [1]. In contrast, acute low back pain is often transient and has a clear underlying cause, such as mechanical injury, while the etiology of CLBP is often multifactorial, encompassing structural, biomechanical, neurological, psychological, and social elements. While the general prognosis is good, the high prevalence rate is what makes CLBP a major public health issue, especially given that CLBP prevalence increases linearly with age [2] and the global population is ageing [3]. In 2020, the number of low back pain cases was estimated to be 619 million (95%, uncertainty interval (UI) 554–694 million), with the projected number of cases rising to 843 million (95%: UI 59–933) by 2050 [4]. Globally pooled, the overall cost, per patient, per annum was estimated at USD 10,100 (95% CI USD 6100–USD 14,200) [5].

According to clinical guidelines, exercise therapy (ET) is broadly recommended as the first line of treatment for reducing pain and disability, with no clear evidence supporting any specific modality over the others [6]. Given the multifactorial nature of CLBP’s causes, as well as the influence of the psychosocial aspects of pain and disability [7], exercise therapy is often paired with various other forms such as pharmacotherapy (paracetamol, nonsteroidal anti-inflammatory drugs), patient education, psychosocial interventions, work hardening, and multidisciplinary rehabilitation. Alternatively, invasive treatments, such as surgery, spinal injections, and radiofrequency denervation, are also used [6,8,9]. An updated overview that collected national clinical practice guidelines identified that 9 out of 11 guidelines recommend multidisciplinary rehabilitation [8].

Multidisciplinary biopsychosocial rehabilitation (MBR) is a comprehensive approach to the management of conditions like chronic low back pain (CLBP). It is rooted in the biopsychosocial model, which posits that biological, psychological, and social factors all play significant roles in human functioning in the context of disease or illness [10]. Key to the MBR approach is that the intervention program should be delivered by a team of healthcare professionals from different backgrounds. This team may include physicians, psychologists, physiotherapists, social workers, occupational therapists, and others. At least two professionals from different backgrounds should be involved in the intervention delivery [11]. It is vital that the various components of the intervention are integrated and that there is active communication between the providers responsible for different aspects of the patient’s care.

While research into MBR as a treatment option for CLBP has shown [10,12] its viability, similarly to ET, there are no recommendations as to what modality of MBR has the largest effect on pain and disability, nor how ET, a first-line treatment, compares with various modalities of MBR. A network meta-analysis, which simultaneously compares multiple treatment modalities, is potentially more effective for comparing treatment options. The research question posed in this systematic review is as follows:

How do the modalities of multidisciplinary biopsychosocial rehabilitation (MBR) compare to exercise therapy (ET), usual care (UC), and minimal intervention (MI) in terms of their efficacy for short-term pain and disability relief in individuals with chronic low back pain?

## 2. Materials and Methods

### 2.1. Protocol and Registration

This meta-analysis was carried out following the PRISMA extension statement for the reporting of systematic reviews incorporating network meta-analyses of health care interventions [13] and is registered with PROSPERO (CRD42022321892) [14].

### 2.2. Eligibility Criteria

The study characteristics adhered to the PICOS framework:

Population: The systematic review focused on adult individuals (18+ years old) with non-specific low back pain persisting for longer than 12 weeks. Studies examining individuals with serious medical conditions that mimic CLBP symptoms such as trauma injury, compressive vertebral fracture, disc herniation, spinal stenosis, rheumatic disease, and cancer were excluded. Conditions like disc degeneration, bulging disc, and osteoarthritis of facet joints were included, given their commonality and often non-severe symptoms. In terms of context, studies in outpatients’ clinics and other clinical settings were included. No context-related exclusion criteria were identified.

Intervention: In the context of this systematic review, MBR was characterized as an intervention that includes a physical component, such as an exercise program or similar physiotherapy intervention, combined with at least one other component drawn from the psychological or social and occupational domains of the biopsychosocial model. We initially considered studies examining four modalities: behavioral (MBR-BE), biofeedback, work/physical conditioning programs (MBR-WR), and education programs (MBR-ED) based on previous systematic reviews [10,15].

Studies where surgical intervention was undertaken at any point before intervention were excluded, but studies that used pharmacotherapy alongside the investigated therapeutic modalities were included.

Comparator: Three comparators were used for the network—exercise therapy (ET) and two controls, namely minimal intervention (MI) or usual care (UC). A comparator was classified as MI if the study stated explicitly that no therapy was provided for the participants. This included participants on waiting lists and participants who were instructed how to generally manage CBLP but were never given any specifics on exercise or how to modify their activities. UC received standard care for CLBP, usually including some form of physiotherapy and/or general exercises for lower back issues. Studies featuring surgical interventions before ET, MI, or UC as comparators were excluded.

Outcome: The main outcomes were pain and disability. Pain was primarily measured via the Visual Analogue Scale, the McGill Pain Questionnaire, and the Numerical Rating Scale, while disability was measured via the Roland–Morris Disability Questionnaire, the Quebec Back Pain Disability Scale, the Oswestry Low Back Pain Disability Questionnaire, the Pain Disability Index, and the Hannover Functional Ability Questionnaire. For inclusion, only outcomes measured immediately after intervention were considered for this study.

Study Design: Only randomized controlled trials (RCT) were included in this systematic review.

### 2.3. Information Sources and Search Strategy

Information sources included the following electronic databases: MEDLINE, PEDro, EMBASE, CINAHL, CENTRAL, and PsycINFO. The literature search included studies up to 4 March 2022. No gray literature, clinical trial registries or regulatory agencies websites were searched. The authors of the included studies were contacted if any clarification or additional data were needed for their studies. The complete search strategy is available in the Appendix A.

### 2.4. Study Selection

Studies were screened by title and abstract independently by two researchers (KD and LE). Any disagreements were resolved by a third researcher (IJ). Full text selection was conducted by the same researchers. Studies were considered eligible if they compared any combination of ET, MBR, and either control (MI or UC). Studies were excluded if the pain experienced by the participants was explicitly stated to be specific and acute. If no such statements were made, we examined the inclusion criteria for specific diagnoses and reviewed the sample description for reported pain duration to determine whether the underlying cause of the pain was nonspecific. Studies were also excluded if they did not measure pain or disability. The review process utilized the Covidence review management system [16].

### 2.5. Data Collection Process and Data Items

Raw data were extracted according to a prepared data extraction form from eligible studies by two researchers (KD and LE). A third researcher (IJ) extracted data that needed recalculation (i.e., standard errors (SE) to standard deviations (SD), or medians to means), or if only the graphically presented data were available. If it was not possible to extract the data, the authors of the original studies were contacted twice to supply their data. The last step was to check whether the extracted data matched the data used in previous similar systematic reviews and meta-analyses [10,17]. As a final measure, due to strong relationships between the means and SDs of all studies, the SDs for two studies with missing data were imputed using a linear regression model.

Along with the intervention groups, mean, SD and sample size, the following variables of interest were extracted with the purpose of inclusion in the analysis: specific measures used for both pain and disability outcome assessment, the number of male and female participants per intervention group, the mean duration of therapeutic intervention in weeks, the mean length of therapeutic interventions in hours per week, the mean age, the mean BMI, and the mean duration of symptoms in months.

### 2.6. Geometry of the Network

The network incorporates multidisciplinary biopsychosocial rehabilitation with the following modalities: behavioral (MBR-BE) and education programs (MBR-ED), work/physical conditioning (MBR-WR), and exercise therapy (ET). Finally, usual care (UC) and minimal intervention (MI) were also included as nodes. Minimal intervention was used as the reference treatment. The geometry of the network is represented in network diagram (Figure 1).

### 2.7. Risk of Bias within Individual Studies and across Studies

Risk of bias was assessed at the study level using Cochrane’s Risk of Bias 2 tool [18]. Studies were categorized by their overall risk of bias into low bias, some concerns and high bias categories. The overall risk of bias in our network meta-analysis was determined by an algorithm that considered the importance and level of bias across different domains. We identified D1, D3, and D4 as the most impactful domains for the bias assessment due to their inherent attributes in study design and data management. The Rob 2 algorithm for overall risk of bias is detailed in the Appendix A. Rob 2 was visualized using the *robvis* tool [19].

The possible presence of publication bias was evaluated using funnel plots and Egger’s test.

### 2.8. Summary Measures

Data collection and analysis were performed based on outcomes and specific data points. Diverse visual analogue scales for pain outcomes were rescaled to 0–100, employing a similar approach to in the most recent Cochrane review on a related topic [17]. As for disability outcomes, multiple scales were also utilized. These Health-Related Quality of Life Questionnaires (HRQoL), which are inherently more complicated than visual analogue scales, were maintained as raw data for the calculation of Hedges’ *g* standardized mean difference (SMD) effect size estimators [20].

Since we were primarily interested in the modalities of MBR, if a particular study had more than one ET modalities, their means and SDs were pooled together according to procedures for pooling groups described in Cochrane’s manual [21].

### 2.9. Planned Methods of Analysis

All statistical analyses were conducted using the R statistical software (version 4.3.1), utilizing the *netmeta* package for network meta-analysis (NMA) [22].

For the analysis, a frequentist random-effects NMA was conducted, and multi-arm studies were integrated into the network. The non-independence of these multi-arm studies was addressed by reweighting all comparisons within each study.

### 2.10. Assessment of Heterogeneity and Inconsistency

Due to high expected heterogeneity, the restricted maximum-likelihood estimator (REML) was used for calculating between-study variance (tau^2^), with a Q-profile for calculating the confidence interval of tau^2^ and tau. Treatment rankings were determined using P-scores, which are the frequentist counterparts of SUCRA values [20]. Global inconsistency was evaluated using the Q statistic based on a full design-by-treatment interaction random effects model (DBT model), whereas local inconsistencies were assessed through node-splitting analysis.

### 2.11. Additional Analyses

Given that we expected high heterogeneity and inconsistency, a Bayesian network meta-analysis, using the same dataset and non-informative priors, employing Markov Chain Monte Carlo simulation with 10^5^ iterations and a burn-in period of 5000 iterations was performed as sensitivity analysis to better assess uncertainty and to perform a network meta-regression analysis using RoB 2 as a covariate. Treatment rankings by surface under the cumulative ranking curve (SUCRA) values were compared to rankings by frequentist *p*-value rankings.

## 3. Results

### 3.1. Study Selection and Included Studies’ Charateristics

The initial databases search identified 4619 references and 4616 studies. Automated duplication screening removed 1836 duplicates. The title and abstract screening phase removed 2308 studies, and 380 additional studies were removed during the full-text screening phase. Two-thirds (66.8%) of the removed studies compared interventions that were of no interest for this systematic review and subsequent meta-analysis. Finally, 93 (87 for pain outcomes and 74 for disability outcomes) studies were marked for data extraction and analysis [23,24,25,26,27,28,29,30,31,32,33,34,35,36,37,38,39,40,41,42,43,44,45,46,47,48,49,50,51,52,53,54,55,56,57,58,59,60,61,62,63,64,65,66,67,68,69,70,71,72,73,74,75,76,77,78,79,80,81,82,83,84,85,86,87,88,89,90,91,92,93,94,95,96,97,98,99,100,101,102,103,104,105,106,107,108,109,110,111,112,113,114,115]. The PRISMA flow diagram can be found in the Appendix A.

For screening, the inter-rater reliability, assessed using Cohen’s kappa, demonstrated moderate agreement in title and abstract screening (κ = 0.542), and fair agreement during the full-text screening phase (κ = 0.357).

Table 1 summarizes the basic parameters of the population in the included studies. In our study, we included a total of 93 trials with a combined participant count of 8059 individuals.

Most of the included studies (84 studies, 96.55%) employed the Visual Analogue Scale (VAS) as the primary instrument to gauge pain outcomes, with values ranging from 0 to 10 or from 0 to 100. A few other scales, like the McGill Pain Questionnaire (MGPQ, 2 studies, 2.3%) and the Pain Rating Chart (PRC, 1 study, 1.15%), were also used. Regarding disability outcomes, the Roland–Morris Disability Questionnaire (RMDQ, 41 studies, 47.13%) and the Oswestry Disability Index (ODI, 30 studies, 39.19%) were the most utilized. Other scales used included the Quebec Back Pain Disability Scale (QBPDS, 3 studies, 4.05%) and the Physical Disability Index (DI, 1 study, 1.35%). A table containing the study characteristics of pain and disability outcomes, comparisons, the number of participants per study, the duration of follow-up, and the setting of the study is available in the Appendix A.

### 3.2. Presentation of Network Structure and Summary of Network Geometry

Figure 1 shows a network structure for pain and disaiblity outcomes, respectively. Blue, light blue and gray tringles indicate the presence of multi-arm studies in these particlular comparisons of nodes. Unsurprisingly, exercise therapy in general, followed by minimal intervention and usual care are the most connected nodes to the network, while MBR modalities have fewer connections. There was only one included study evaluating biofeedback and it was merged into the behavioral MBR group (MBR-BE), as these modalities are more closely related than the others.

### 3.3. Risk of Bias within Studies

A summary of the risk of bias, using Risk of Bias 2 tool, is presented in Figure 2. In evaluating the risk of bias in our collected studies on chronic low back pain treatment, we observed variability in quality. The risk of bias arising from the randomization process was generally low or of some concern, indicating satisfactory randomization procedures. Bias due to deviations from intended interventions varied, with some studies being flagged for high bias, suggesting potential implementation discrepancies. Most studies maintained solid data integrity, indicated by the predominantly low bias due to missing outcome data. Conversely, we noted substantial variations in bias in the measurement of outcomes and in the selection of reported results, with several studies presenting high bias. Ultimately, the overall bias varied broadly across studies. These variances underscore the necessity for caution in the rigorous interpretation of analysis. A figure depicting the complete risk of bias for the included studies can be found in the Appendix A.

### 3.4. Results of Comparions and Synthesis of Results

The league table (Table 2) summarizes the comparative effects of different interventions on pain outcomes, with minimal intervention (MI) serving as the reference therapy. In this table, the lower triangle represents the network meta-analysis (NMA) estimates, which integrate both direct and indirect evidence to provide comprehensive comparisons between interventions. Conversely, the upper triangle showcases direct comparisons derived solely from head-to-head trials between the specific interventions. A positive mean difference indicates that the row intervention is more effective than the column intervention by the stated amount.

Looking at NMA estimations, MBR education (ED) demonstrated the largest mean difference in reducing pain outcomes, with a value of 18.18 (95% CI: 13.06 to 23.30). This was closely followed by MBR behavioral (BE), with a mean difference of 16.96 (95% CI: 10.47 to 23.46). MBR work conditioning/hardening (WR) showed a mean difference of 12.72 (95% CI: 2.05 to 23.39) in pain reduction, while exercise therapy (ET) exhibited a mean difference of 12.37 (95% CI: 8.40 to 16.34). It is noteworthy that some comparisons included negative lower limits in their confidence intervals, indicating a nonsignificant effect. A table of the effect sizes and standard errors of pain and disability outcomes by study is available in the Appendix A.

Table 3 shows the direct comparison and NMA estimates concerning disability outcomes across varying interventions. A positive, higher value of the standardized mean difference indicates that the row intervention is more effective than the column intervention by the stated amount. Looking at indirect estimations and using minimal intervention (MI) as the reference, MBR behavioral (MBR-BE) exhibited the largest SMD of 0.88 (95% CI 0.46 to 1.30), with no direct comparison available.

MBR education (MBR-ED) displayed a significant reduction in disability with a standardized mean difference (SMD) of 0.67 (95% CI: 0.40; 0.94), while exercise therapy (ET) showed a relatively similar efficacy with an SMD of 0.53 (95% CI: 0.32; 0.74) in comparison to MI. The range of the associated 95% CIs underscores the variability and uncertainty in some of these estimates, making it crucial to interpret these results with caution.

In the assessment of pain and disability outcomes, the treatments ranked based on their probabilities that a particular treatment is better than another treatment chosen at random (P-scores) revealed distinct hierarchies (Table 4). For pain outcomes, the MBR-ED took precedence, with the highest P-score of 0.899, followed closely by MBR-BE at 0.826 and MBR-WR at 0.559. Parallel trends were observed in disability outcomes. MBR-BE led with a P-score of 0.940, MBR-ED was second, at 0.761.

The modalities ranked first and second for pain outcomes have very similar probabilities, which decline more sharply for the third-ranked modality in both outcomes. While examining rankings provides an approximation of the most useful therapeutic modality, it does not paint the full picture. Although the calculation of P-scores accounts for variance, visually inspecting a forest plot with effect size estimates against a common reference therapy reveals the confidence surrounding the point estimate, and consequently the confidence in the rankings. Figure 3 summarizes MD/SMD for pain and disability when compared to minimal intervention. From the figure, for pain outcomes, MBR-ED appears to be the most successful, with MBR-BE as a close runner-up. Evaluating the SMDs and 95% CIs for disability outcomes, the first three modalities, MBR-BE, MBR-ED, and ET, are all quite similar. While MBR-BE has the largest effect size, it is reasonably uncertain as to which one is truly the most effective.

### 3.5. Exploration for Heterogeneity and Inconsistency

For the network assessing pain outcomes, the total Q statistic of 920.85 with 85 degrees of freedom (df) signifies considerable heterogeneity within the network (*p* < 0.001). This heterogeneity can be partitioned into two parts: within-designs and between-designs. Significant heterogeneity was observed both within designs (Q = 628.07, df = 74, *p* < 0.001) and between designs (Q = 292.78, df = 11, *p* < 0.001). Under the assumption of a full design-by-treatment interaction random effects model, the Q statistic was non-significant (Q = 14.39, df = 11, *p* = 0.2124), suggesting no notable inconsistency between designs. Local inconsistency assessments highlighted two significant inconsistencies. Specifically, the comparisons ‘exercise therapy vs. MBR education’ and ‘exercise therapy vs. minimal intervention’ were found to be inconsistent with *p*-values of 0.044 and 0.046, respectively. Other therapeutic comparisons did not demonstrate significant inconsistency.

For the network assessing disability outcomes, significant heterogeneity was present, as indicated by a total Q statistic of 316.89 (df = 70, *p* < 0.001). The heterogeneity within designs was substantial (Q = 282.52, df = 65, *p* < 0.001), and between designs, it was also significant (Q = 34.37, df = 5, *p* < 0.001). Under the presumption of a full design-by-treatment interaction random effects model, the Q statistic for between designs was 9.53 (df = 5, *p* = 0.090), indicating a possible inconsistency. When focusing on specific comparisons, three stood out. The comparisons of ‘exercise therapy vs. minimal intervention’, ‘MBR education vs. minimal intervention’, and ‘exercise therapy vs. MBR education’ were statistically significant with *p*-values of 0.021, 0.008, and 0.045, respectively. All other comparisons were non-significant.

Comprehensive details regarding the heterogeneity and inconsistency of both network meta-analysis models are provided in the Appendix A.

### 3.6. Risk of Bias across Studies

For pain outcomes, Egger’s test showed no statistically significant results (*p* = 0.124). Visual inspection of the plot suggested a potential asymmetry, indicating the possibility that studies demonstrating non-positive effects of the experimental therapeutic modalities as opposed to minimal intervention may be underrepresented in the published literature. While it is important to note that Egger’s test, despite its widespread use, is known to have low power and can fail to detect bias [116], funnel plot shows no significant symmetry deviation.

For disability outcomes both Egger’s test (*p* = 0.003) and visual assessment of funnel plot suggest a possibility of publication bias. Funnel plots for both outcomes are available in the Appendix A.

### 3.7. Results of Additional Analyses

The results of the Bayesian network meta-analysis (BNMA), employing a non-informative prior distribution, largely corroborated the findings from the frequentist network meta-analysis (NMA). For pain-related outcomes, node-splitting analysis revealed no statistically significant inconsistencies among the comparisons. Notably, the therapeutic modalities were ranked according to SUCRA scores in a manner consistent with the frequentist ranking.

Similar outcomes were observed for disability-related endpoints. As in the frequentist NMA, the results exhibited substantial consistency, with the only inconsistency identified occurring in the comparisons between exercise therapy (ET) and minimal intervention (MI), as well as between ET and MBR-ED (MBR education) and between MBR-ED and MI.

Within the Bayesian framework, we conducted a network meta-regression employing the RoB 2 Overall domain as a moderator. Interestingly, this model did not provide any meaningful explanations for inconsistencies within the network. Furthermore, there were no significant differences in effect sizes when considering the grouping of studies into ‘low risk’ and ‘some concerns,’ versus ‘high risk’ categories for both pain and disability outcomes. However, it is worth noting that effect sizes generally appeared more substantial in ‘low-risk’ studies. For a comprehensive BNMA analysis of both outcomes, refer to the Appendix A.

## 4. Discussion

The aim of the research was to discern the efficacy of multidisciplinary biopsychosocial rehabilitation (MBR) modalities relative to exercise therapy (ET) for short-term pain and disability relief in chronic low back pain sufferers.

Our results support the previous research in regard to the efficacy of ET and MBR modalities versus minimal intervention and usual care. The efficacy of ET versus minimal intervention or usual care is well established [17]. The MBR approach has less evidence to support its efficacy, but previous studies do show promise when compared to usual care [12] or exercise therapy [10]. When indirectly compared, a hierarchy emerged in the P-scores of the pain outcomes, with MBR-ED leading, followed by MBR-BE. Parallel to the findings on pain outcomes, the P-score ranking for disability outcomes positioned MBR-BE at the top rank, followed by MBR-ED. Both MBR-ED and MBR-BE showed larger effects on pain and disability than ET and MBR-WR, which in turn, as expected, showed larger effects than UC and MI. This demonstrates that better outcomes are associated with modalities that have enhanced physiotherapy interventions with a cognitive aspect, either through behavioral or educational modalities.

This should be examined through the multifactorial nature of chronic low back pain (CLBP). A recent systematic review [117] found that increased pain intensity, elevated body weight, lifting heavy objects at work, challenging work postures, and depression are the most commonly observed risk predictors for CLBP. Additionally, behaviors that are not adaptive, general anxiety, functional limitations during the episode, and particularly physically demanding work are distinctly linked to the persistence of symptoms. The most frequently identified protective factor against CLBP was regular physical activity. Given this variety of risk factors, it stands to reason that effectively addressing CLBP requires a multifaceted approach, as suggested by our results.

Defining which of these MBR approaches is superior is much more challenging. To our knowledge, there have been no previous attempts to try to assess different types of MBR in comparison with each other. While the results produced rankings, giving MBR-ED and MBR-BE the highest rank of pain and disability outcomes, respectively, there is a high degree of uncertainty within the results, evidenced by the wide confidence intervals, significant heterogeneity, and some local inconsistency within the network. This was generally anticipated, given the similar findings in other studies [10,12,17] and because, by design, studies were broadly included and categorized as being relevant to clinicians, researchers, and policymakers, but such high heterogeneity is a deterrent to producing confident results. Even though ranking treatments favored MBR-ED for pain and MBR-BE for disability outcomes, the differences between the various MBR modalities are too minimal and uncertain to be clinically significant. Furthermore, for disability outcomes, ET cannot be reliably differentiated from its MBR counterparts.

Regarding the risk of bias within the individual studies, the integrity of randomization in clinical trials is safeguarded by concealing the allocation sequence, which prevents selection bias and ensures comparability between intervention groups. In our systematic review, we found that a substantial number of studies (58 out of the total) explicitly mentioned that the random allocation was adequately concealed, and confirmed the absence of baseline imbalances, which speaks to the methodological robustness of these trials (RoB2 item 1.2). Like allocation concealment, blinding within clinical trials serves as a cornerstone in preventing several types of biases, including performance and detection biases. Within the context of rehabilitation research, the nature of the interventions often precludes the blinding of participants and caregivers, particularly when interventions such as exercise therapy, education, and behavioral treatments require active participant cooperation. In our review, only 15 studies were considered as being at a ‘low risk’ of bias due to the blinding of participants and personnel (RoB2 items 2.1 and 2.2), while 40 studies presented ‘some concerns.’ Furthermore, our analysis employed a Bayesian network meta-regression framework, which allowed for a more nuanced exploration of the potential impact of study quality on our results. Although more than 50% studies were judged to be at a high risk of bias by the overall assessment, and despite the intuitive expectation that risk of bias would significantly influence effect sizes, our findings did not reveal a statistically significant relationship in this regard. This was observed across both pain and disability outcomes, even when categorizing studies based on their risk of bias. These findings suggest that while risk of bias is a crucial factor in assessing the quality of evidence, its direct impact on the effect sizes in the context of our specific research may be more complex than traditionally perceived. The Bayesian approach provided a robust platform for this analysis, reinforcing the conclusions drawn from our frequentist network meta-analysis. The consistency in results between the Bayesian and frequentist methods, despite the nuanced differences in their analytical approaches, adds further credibility to our findings. This aspect of our analysis highlights the importance of considering multiple analytical perspectives in systematic reviews and meta-analyses, particularly when dealing with multifaceted and variable interventions like those examined in our study.

### 4.1. Limitations

High heterogeneity, especially within designs, was present. Although consistency assessed under the assumption of a full design-by-treatment interaction random effects model was non-significant, three comparisons also showed local inconsistency. Employing the RoB 2 Overall domain as a moderator in the Bayesian framework, however, did not yield any significant results in explaining network heterogeneity and inconsistency. Regarding the exploration of heterogeneity, the lack of sample characteristics data in many studies, such as age, male to female ratio, BMI, duration of symptoms, and use of medications, prevented us from exploring the effects of study modifiers.

Additionally, in this analysis we did not analyze different forms of ET, which may help to narrow down the effect modifiers and differences in the effect sizes.

There is some evidence of publication bias in disability outcomes. The results of Egger’s test for pain outcomes were non-significant, the test’s low power warrants caution in its interpretation, and the potential underrepresentation of studies showing non-positive effects compared to minimal intervention must be considered.

Furthermore, the inclusion of a diverse range of scales to measure disability outcomes introduces challenges in direct comparisons and synthesis of results.

Finally, like other systematic reviews, the undocumented use of pain medications might have influenced the results of the primary studies, thus making our estimates potentially biased. However, due to randomization in primary studies, the use of medication may be assumed to be comparable between study arms.

### 4.2. Recommendations for Stakeholders

Clinicians: Recognizing the multifaceted etiology of CLBP, while exercise therapy (ET) remains a mainstay, it might be prudent to augment ET with educational sessions. These sessions could elucidate potential pain triggers, preventive measures, and coping strategies for disability. Additionally, clinicians should remain attuned to patients’ psychological well-being, and, when deemed necessary, consider referrals for behavioral interventions to broadly address CLBP’s dual physical and emotional facets.

Researchers: The strength of network meta-analysis (NMA) lies in its ability to discern comparative effectiveness across a spectrum of treatments. However, ensuring that the results remain unbiased mandates careful attention to potential sources of heterogeneity and inconsistency within the network. Delving deeper into patient characteristics like age, gender distribution, BMI, clinical setting, intervention duration, and intensity could shed light on these disparities. Moreover, recognizing the intricacies of CLBP, it is advisable to blend various interventions, encompassing educational components, behavioral modifications, and pain management tactics with ET in experimental designs to discern the optimal therapeutic combination.

Policy-Makers: CLBP’s impact transcends individual suffering, translating into significant socio-economic ramifications. Notably, MBR modalities, by virtue of involving multiple professionals, inherently command higher costs. While preliminary evidence suggests a potential edge in efficacy over ET, it would be counterintuitive to further compound the financial burdens associated with CLBP by advocating for costlier treatments without discernible clinical superiority. Hence, a thorough cost–benefit analysis is warranted to ascertain the most economically viable and clinically efficacious treatment modalities, potentially integrating salient components of MBR into standard ET.

## Figures and Tables

**Figure 1 jcm-12-07489-f001:**
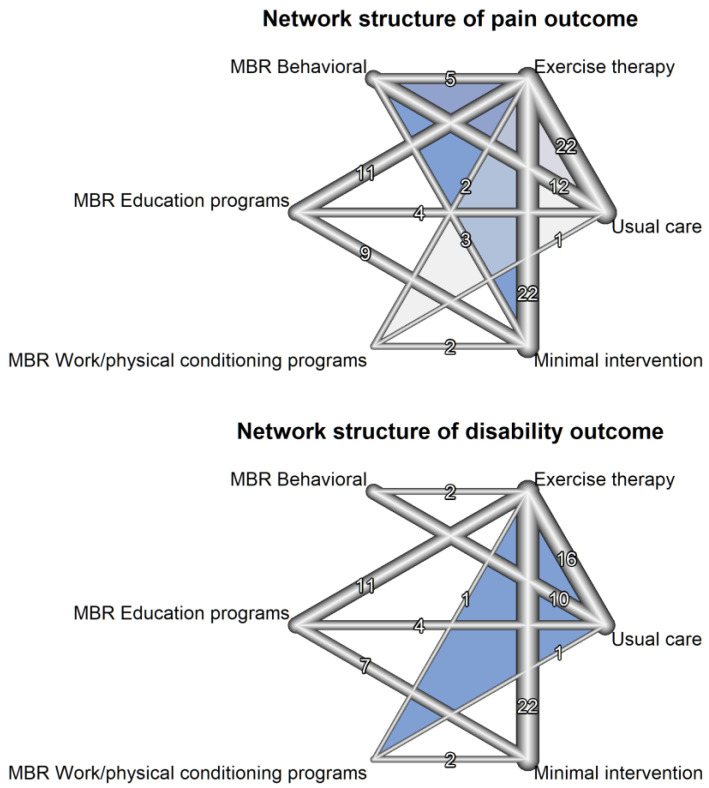
Network structure of pain and disability outcomes (The number between nodes represent number of studies for that particular comparison).

**Figure 2 jcm-12-07489-f002:**
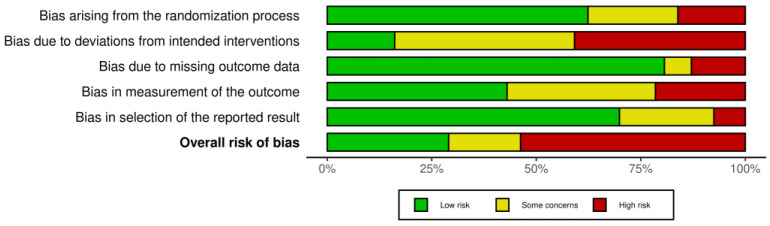
Summary of the risk of bias of included studies.

**Figure 3 jcm-12-07489-f003:**
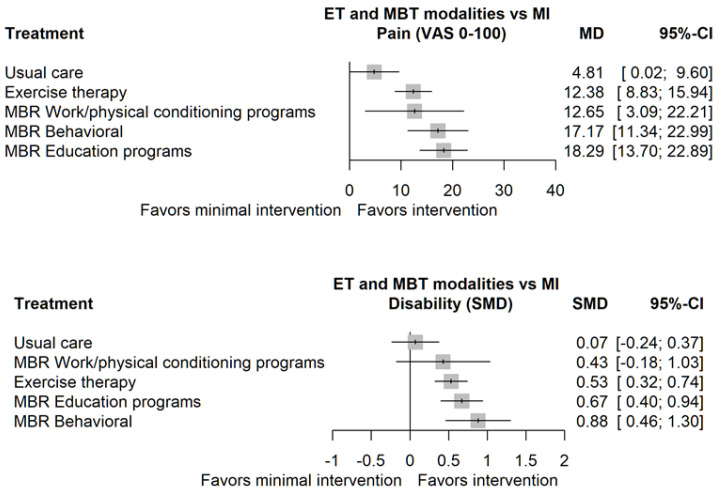
Exercise therapy and MBR modalities vs. minimal intervetion in pain outcomes.

**Table 1 jcm-12-07489-t001:** Characteristics of included trials population.

Population Characteristics	Mean (Min, Max)	Studies	Sample Size
Male (%)	31.18%	72	6476
Age (years)	44.61 (21.4 to 73.63)	88	7432
BMI	25.89 (20.77 to 35)	61	5075
Symptom duration (months)	53.6 (5.3 to 222)	40	4101
Intervention duration (weeks)	9.01 (1 to 24)	84	7163
Number of hours per week	2.3 (0.12 to 30)	80	6867

**Table 2 jcm-12-07489-t002:** League table with direct comparison and NMA estimates of pain outcomes.

**MI**	.	14.56 (10.41; 18.71)	14.03 (−0.59; 28.65)	9.96 (−1.68; 21.59)	14.06 (7.43; 20.68)
4.81 (0.02; 9.60)	**UC**	9.08 (4.90; 13.27)	0.20 (−17.86; 18.26)	11.43 (5.59; 17.28)	11.26 (1.62; 20.90)
12.38 (8.83; 15.94)	7.57 (3.95; 11.20)	**ET**	1.53 (11.59; 14.64)	10.27 (1.78; 18.77)	9.99 (4.17; 15.80)
12.65 (3.09; 22.21)	7.84 (−1.99; 17.68)	0.27 (−9.19; 9.73)	**MBR-WR**	.	.
17.17 (11.34; 22.99)	12.36 (7.52; 17.20)	4.79 (−0.36; 9.93)	4.52 (−6.02; 15.05)	**MBR-BE**	.
18.29 (13.70; 22.89)	13.49 (8.42; 18.55)	5.91 (1.67; 10.16)	5.64 (−4.49; 15.77)	1.13 (−5.19; 7.44)	**MBR-ED**

(values: mean difference (95% CI)). **MI** = minimal intervention; **UC** = usual care; **ET** = exercise therapy; **MBR-WR** = MBR work conditioning/hardening; **MBR**-**BE** = MBR behavioral; **MBR-ED** = MBR education.

**Table 3 jcm-12-07489-t003:** League table with direct comparison and NMA estimates of disability outcomes.

**MI**	.	0.55 (−0.24; 1.35)	0.66 (0.42; 0.89)	0.28 (−0.12; 0.68)	.
0.07 (−0.24; 0.37)	**UC**	−0.11 (−1.15; 0.92)	0.44 (0.17; 0.71)	0.73 (0.17; 1.30)	0.81 (0.46; 1.16)
0.43 (−0.18; 1.03)	0.36 (−0.28; 0.99)	**MBR-WR**	−0.05 (−1.10; 0.99)	.	.
0.53 (0.32; 0.74)	0.46 (0.23; 0.70)	0.10 (−0.51; 0.72)	**ET**	0.36 (0.03; 0.69)	0.37 (−0.36; 1.10)
0.67 (0.40; 0.94)	0.60 (0.29; 0.91)	0.24 (−0.40; 0.89)	0.14 (−0.11; 0.39)	**MBR-ED**	.
0.88 (0.46; 1.30)	0.81 (0.49; 1.13)	0.45 (−0.24; 1.15)	0.35 (−0.02; 0.72)	0.21 (−0.21; 0.64)	**MBR-BE**

(values: standardized mean difference (95% CI)). **MI** = minimal intervention; **UC** = usual care; **ET** = exercise therapy; **MBR-WR** = MBR work conditioning/hardening; **MBR-BE** = MBR behavioral therapy; **MBR-ED** = MBR education.

**Table 4 jcm-12-07489-t004:** Rankings of P-score scores for pain and disability outcomes.

Pain Outcome			Disability Outcome		
Rank	Treatment	P-Score	P-Score	Treatment	Rank
1	MBR-ED	0.899	0.940	MBR-BE	1
2	MBR-BE	0.826	0.761	MBR-ED	2
3	MBR-WR	0.559	0.559	ET	3
4	ET	0.503	0.496	MBR-WR	4
5	UC	0.207	0.161	UC	5
6	MI	0.006	0.082	MI	6

**MI** = minimal intervention; **UC** = usual care; **ET** = exercise therapy; **MBR-WR** = MBR work conditioning/hardening; **MBR-BE** = MBR behavioral therapy; **MBR-ED** = MBR education.

## Data Availability

The data presented in this study are available on request from the corresponding author. The data are not publicly available as they contain additional information that will be used in subsequent analyses for future publications.

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
