# Peer review of "Effects of Multidisciplinary Biopsychosocial Rehabilitation on Short-Term Pain and Disability in Chronic Low Back Pain: A Systematic Review with Network Meta-Analysis"

_jcm, 2023, doi:10.3390/jcm12237489_

Round 1

Reviewer 1 Report

Comments and Suggestions for Authors

This review was very difficult to find a final conclusion because the studies were different. It is difficult to watch a difference between the different therapy ways.

Therefore the study design was not well structurated, it was a high heterogeneity, undocumented use of pain medications might have influenced the results of the primary studies, the study idea is relevant in the daily practice,

The methods are structured , the references well choosen, and discussion too short

Reviewer 2 Report

Comments and Suggestions for Authors

The article is very interesting and contributive. However, it is need to revise.

1- In search strategy mention to "Biofeedback", in the event that "Biofeedback" is MESH TERM. if you mean "Biofeedback, Psychology"[Mesh], please correct your search syntax.

2- This is also true for "Patient Education as Topic"[Mesh] instead of "Patient Education".

3- In my opinion, its better use a table for present of search strategies.

4- Why you haven't use of gray literature?

5- It is appropriate to mention the following items in sup-table1:

- Duration of follow-up

- Exact type of intervention

- Description of setting

6- Have the individual studies mentioned hiding the random allocation list and its solutions?

7- Have the individual studies mentioned blindness?

8-Was there a relationship between the methodological quality (risk bias) of the primary studies and their outcome on the effect size scale? It would have been better to analyze this result.

9-It is better to mention the risk of bias of primary studies in the discussion or conclusion.

Wishing your team the best of success

Round 2

Reviewer 1 Report

Comments and Suggestions for Authors

The authers did the changes in the correct way